# The Impact of Liquid Biopsy in Advanced Ovarian Cancer Care

**DOI:** 10.3390/diagnostics14171868

**Published:** 2024-08-26

**Authors:** Antoni Llueca, Sarai Canete-Mota, Anna Jaureguí, Manuela Barneo, Maria Victoria Ibañez, Alexander Neef, Enrique Ochoa, Sarai Tomas-Perez, Josep Mari-Alexandre, Juan Gilabert-Estelles, Anna Serra, Maria Teresa Climent, Carla Bellido, Nuria Ruiz, Blanca Segarra-Vidal, Maria Llueca

**Affiliations:** 1Reference Unit of Abdominal Pelvic Oncology Surgery (RUAPOS), General University Hospital of Castellón, 12004 Castellón, Spain; sarai.canete.mota@gmail.com (S.C.-M.); serraa@uji.es (A.S.); maclimen@uji.es (M.T.C.); bellido_mca@gva.es (C.B.); nruiz@uji.es (N.R.); 2Oncological Surgery Research Group (OSRG), Department of Medicine, University Jaume I (UJI), 12071 Castellon, Spain; ajauregu@uji.es (A.J.); barneo@uji.es (M.B.); neef@uji.es (A.N.); 3Department of Mathematics, IMAC (Institut Universitari de Matematiques i Aplicacions de Castelló), University Jaume I (UJI), 12071 Castellon, Spain; mibanez@uji.es; 4Department of Molecular Biology, Hospital Provincial de Castellon, 12002 Castellón, Spain; enrique.ochoa@hospitalprovincial.es; 5Research Laboratory in Biomarkers in Reproduction, Gynaecology and Obstetrics, Research Foundation of the General University Hospital of Valencia, 46014 Valencia, Spain; sarai.altea@gmail.com (S.T.-P.); josepmarialexandre@gmail.com (J.M.-A.); juangilaeste@yahoo.es (J.G.-E.); 6Pathology Department, General University Hospital of Valencia Consortium, 46014 Valencia, Spain; 7Department of Medical Oncology, Hospital Provincial de Castellon, 12002 Castellón, Spain; 8Gynecology Oncology, La Fe University and Polytechnic Hospital, 46026 Valencia, Spain; blancasv@icloud.com; 9Department of Obstetrics and Gynecology, Joan XXIII University Hospital of Tarragona, 43005 Tarragona, Spain; mllueca.hj23.ics@gencat.cat

**Keywords:** advanced ovarian cancer, liquid biopsy, translational research, cancer care

## Abstract

Introduction: Ovarian cancer is the third most common gynaecological cancer and has a very high mortality rate. The cornerstone of treatment is complete debulking surgery plus chemotherapy. Even with treatment, 80% of patients have a recurrence. Circulating tumour DNA (ctDNA) has been shown to be useful in the control and follow-up of some tumours. It could be an option to define complete cytoreduction and for the early diagnosis of recurrence. Objective: We aimed to demonstrate the usefulness of ctDNA and cell-free DNA (cfDNA) as a marker of complete cytoreduction and during follow-up in patients with advanced ovarian cancer. Material and Methods: We selected 22 women diagnosed with advanced high-grade serous ovarian cancer, of which only 4 had complete records. We detected cfDNA by polymerase chain reaction (PCR), presented as ng/mL, and detected ctDNA with droplet digital PCR (ddPCR). We calculated Pearson correlation coefficients to evaluate correlations among cfDNA, ctDNA, and cancer antigen 125 (CA125), a biomarker. Results: The results obtained in the evaluation of cfDNA and ctDNA and their correlation with tumour markers and the radiology of patients with complete follow-up show disease progression during the disease, stable disease, or signs of recurrence. cfDNA and ctDNA correlated significantly with CA125. Following cfDNA and ctDNA over time indicated a recurrence several months earlier than computed tomography and CA125 changes. Conclusion: An analysis of cfDNA and ctDNA offers a non-invasive clinical tool for monitoring the primary tumour to establish a complete cytoreduction and to diagnose recurrence early.

## 1. Introduction

Ovarian cancer is the third most common gynaecological cancer and has a very high mortality rate [1,2]. The diagnosis of ovarian cancer is often delayed, with most patients (70–80%) being diagnosed with advanced stages of the disease. The 5-year survival rate in these patients is <20% [3] Advanced ovarian cancer surgery aims to achieve maximal cytoreduction (absence of macroscopic residual tumour) to increase survival and to provide a definitive cure in some cases. However, the definition of complete cytoreduction (microscopic disease after surgery) is completely subjective [3]. Recurrence is observed in almost 25% of patients with early-stage disease and in >80% of patients with more advance stages. Based on a platinum-free interval cut-off of 6 months, the first recurrence is usually classified in platinum-sensitive versus platinum-resistant tumours, reflecting the biological characteristics underlying the clinical behaviour [4,5]. Thus, the treatment of recurrent ovarian cancer is a significant clinical challenge [1].

Cancer antigen 125 (CA125) is a serum biomarker for ovarian cancer; an increase in its level is an indicator of cancer growth. This test is widely used to monitor patients with ovarian cancer, in the diagnosis, for patients who are receiving therapy, and during the follow-up [6]. Serum CA125 levels are elevated in 50% of early-stage tumours, which are mostly type I ovarian cancer, and in 92% of advanced-stage tumours [7]. A small group of patients maintains normal CA125 levels and thus have no options for early detection of recurrence. Computed tomography (CT) is the current standard for monitoring the response to therapy in patients with advanced cancer and during follow-up. Nevertheless, an increase in one or more tumour markers usually precedes the diagnosis of clinical recurrence based on imaging by a median of 4.5 months (range: 0.5–29.5 months) [8].

Disease progression is defined after complete clinical and radiographic responses with normalisation of CA125 serum levels (≤35 U/mL). Recurrence is established based on the Response Evaluation Criteria in Solid Tumours (RECIST) [9] CA125 levels, namely at least a twofold elevation of CA125 above the upper limit of normal (35 U/mL) on two occasions. Circulating tumour DNA (ctDNA) has been shown to be useful in the control and follow-up of some tumours. ctDNA is released into the bloodstream by apoptotic or necrotic tumour cells [10]. Multiple studies have demonstrated that individuals with cancer have elevated ctDNA levels. This elevation may precede changes that are detected with other established methods, including elevated biomarker serum levels or radiological evidence of disease progression [11,12,13,14,15]. For this reason, the use of ctDNA can be a follow-up option to diagnose recurrence early. Another possible advantage associated with the use of ctDNA is the possibility to relate the value obtained to the residual volume after surgical treatment, one of the main limitations of complete cytoreduction surgery [16,17].

We developed and tested a quantitative polymerase chain reaction (qPCR) assay to monitor cell-free DNA (cfDNA) plasma levels longitudinally in patients with advanced cancer while they receive treatment and during their follow-up. cfDNA in plasma is liberated from different cellular decay processes, both physiologically and pathologically. In addition, we can detect and quantify gene mutations in plasma by using droplet digital polymerase chain reaction (ddPCR). This approach provides a way to assess cancer-related chromosomal instability over time. The objectives of this pilot study were to determine the feasibility of using qPCR and single-color ddPCR to monitor plasma cfDNA and ctDNA trends, respectively, to define complete cytoreduction and to diagnose recurrence earlier than other traditional techniques, including imaging and biomarkers.

## 2. Materials and Methods

### 2.1. Study Design and Patient Characteristics

We selected 22 women diagnosed with advanced high-grade serous ovarian cancer (HGSOC) (stages IIIC–IV according to the International Federation of Gynaecology and Obstetrics [FIGO] 2014 staging system) [18] treated at the Reference Unit in Abdomino-Pelvic Oncological Surgery (UR-COAP) of the Valencian Region (Spain). All procedures were carried out by the same clinical team together with the lab teams of the Castellón General Hospital Consortium and General University Hospital Consortium of Valencia [16,17].

Tissue and plasma were stored at the Biobank for Biomedical and Public Health Research of the Valencian Community (IBSP-CV). Each patient provided informed consent. The study was approved by the Institutional Review Board of our institution and performed according to the ethical principles of the Declaration of Helsinki [19]. Patients with a previous malignancy, synchronous tumours, or who could not be followed up were excluded. 

The preoperative study included a baseline determination of CA125 and a CT scan with determination of the peritoneal carcinomatosis index (PCI) prior to cytoreduction surgery. Patients who were not candidates for primary surgery due to the extent of the disease or their baseline condition were treated with neoadjuvant chemotherapy. Recurrence was diagnosed according to RECIST [9] based on CT imaging and CA125. Chemotherapy was individualised for each patient; it included targeted inhibitor biotherapy, a vascular endothelial growth factor antibody (e.g., bevacizumab), or first- or second-line combinations of chemotherapy. 

For each patient, whole blood samples (10 mL) were collected on a fixed schedule: the date of the diagnostic laparoscopy (t−1); the date of debulking surgery (considered day 0; t0); 24–48 h after the surgery (t1); and 1, 6, 9, 12, and 24 months after surgery (t2, t3, t4, t5, and t6, respectively). The number of total time points sampled for each patient is listed in Table 1. Tumour tissue was collected at t0 or t−1 for diagnostic purposes.

### 2.2. Genomic DNA and cfDNA Isolation

Samples were collected between 8 months before surgery and during 33 months of follow-up after surgery (Table 1). cfDNA was isolated from a total of 55 samples representing these different time points. cfDNA was detected by PCR and quantified in ng/mL, whereas ctDNA was determined by droplet digital PCR (ddPCR). The minor allele frequency (MAF) quantifies the number of mutant and wild-type copies isolated per millilitre of plasma at each time point. The CA125 serum levels were determined for each patient (U/mL). 

DNA was extracted from a single 5 µm formalin-fixed paraffin-embedded (FFPE) tissue section obtained after the debulking procedure with the AllPrep DNA/RNA FFPE (Qiagen, Hilden, Germany), following the manufacturer’s instructions. cfDNA was extracted from 4–6 mL of plasma using the QIAamp Circulating Nucleic Acid Kit (Qiagen, Hilden, Germany) and eluted in 20 μL of AVE elution buffer. The Qubit dsDNA HS Assay Kit and a Qubit 3.0 fluorometer (Invitrogen, Carlsbad, CA, USA) were used to measure the DNA concentrations.

### 2.3. Sequencing

FFPE DNA obtained from the tumour tissue was sequenced on an Ion Torrent S5 sequencer using the tp53_BRCA Ion AmpliSeq™ Custom panels (IAD207963, ThermoFisher Scientific, Carlsbad, CA, USA). It included *TP53*, *BRCA1,* and *BRCA2* with amplicons of 125–140 base pairs (bps) in size and a total target size of 21.8 kilobases (kbs). The majority of mutations in HGSOC are associated with these three genes. Consumables, kits, software packages, and the protocol for the next-generation sequencing (NGS) analyses were used as indicated by the manufacturers. Sequencing resulted in an average read depth of 20,000 (range: 12,000–34,500). Variants found to be of interest were Sanger-sequenced for confirmation. Primers were designed with Primer3 (https://www.primer3.org accessed on 1 January 2021) over the genomic sequences of the respective gene to acquire amplicons of size ranging from 150 to 250 bps. For PCRs, 1 µL of FFPE-extracted DNA, the Applied Biosystems^TM^ AmpliTaq Gold^TM^ Master Mix (ThermoFisher Scientific, Vilnius, VA, Lithuania) and the corresponding forward- and reverse-designed primers (10 µM) were used. PCRs were performed using the following thermal cycling conditions: an initial denaturation at 95 °C for 10 min, followed by 45 cycles of denaturation [94 °C, 45 s], annealing [temperature-dependent primer melting temperature, 45 s], and extension [72 °C, 45 s], and a final extension at 72 °C for 5 mis Obtained amplicons were checked for correct size in 2.5% agarose gels at 90 V. PCR products were cleaned up with exonuclease I (Takara, Sumilab, Kusatsu, Shiga, Japan) and shrimp alkaline phospathase (Takara, Sumilab, Kusatsu, Shiga, Japan) in two steps: incubation [37 °C, 60 min] and inactivation [95 °C, 5 min]. Next, 2.5 µL of clean products was sequenced in both, forward and reverse, directions using the BigDye^TM^ Terminator v3.1 Cycle Sequencing Kit (Applied Biosystems^TM^, ThermoFisher Scientific, Vilnius, VA, Lithuania) and forward or reverse primer (10 µM). The thermal cycling conditions were initial denaturation at 95 °C for 1 min, followed by 25 cycles of denaturation [90 °C, 30 s], annealing [temperature-dependent primer melting temperature, 15 s], and extension [60 °C, 2 min]. Sequencing clean-up was performed with the Performa Dye Terminator Removal Gel Filtration Cartridges (Edge Biosystems Inc, ThermoFisher Scientific, Vilnius, VA, Lithuania) according to manufacturer’s instructions. For PCR amplification, PCR product clean-up, and cycle sequencing, a Mastercycler^®^ ep thermal cycler (Eppendorf) was used. Finally, capillary electrophoresis of marked samples was carried out in the genomics section of the central support service for experimental research of the University of Valencia. Chromatograms were manually aligned to the NGS reads to check for confirmation of the mutation.

### 2.4. Bioinformatic Analysis for Mutation Detection and Evaluation

Sequences were analysed with Ion Reporter^TM^ Software (Version 5.20), which is built into the Ion Torrent sequencer (Ion GeneStudio S5 Systems, Ion Torrent, ThermoFisherScientific, Carlsbad, CA, USA). Variants were called out and read separately using the somatic and germline options. Then, the two parts were joined and the variants were classified according to their MAF. Mutations with an MAF > 45% are more likely to be germline mutations; all others are more likely to be somatic. For this study, because the sequence analysis using NGS was not performed with germline cells, only variants with an MAF between 5% and 45% were considered, to ensure that the mutation found was only in somatic cells [17]. These mutations are also classified as pathogenic in the databases. All available information regarding the pathogenicity of the observed mutations was mined from three databases: ClinVar (https://www.ncbi.nlm.nih.gov/clinvar), VarSome (https://varsome.com), and Franklin (https://franklin.genoox.com/clinical-db/home).

### 2.5. Primer and Probe Design for ddPCR

Each mutation detection assay contained a single set of primers and two competitive probes, one to detect the WT allele (HEX) and one to detect the mutant allele (FAM), with PCR products of 65–70 bps in length. The BIO-RAD (BioRad Laboratories, Hercules, CA, USA ^TM^) company’s mutation detection assays can be validated or designed on request by the researcher (customised). They are unique identification assays that you can request from BIO-RAD.

The company does not provide the primer and probe sequences under any circumstances.

The assays requested for this study are five, the first three validated and the last two not. They are the following:dHsaMDV2010105 for the TP53R175H mutation;dHsaMDV2010127 for the TP53R248Q mutation;dHsaMDV2510542 for the TP53G245D mutation;dHsaMDS99673883 for the BRCA1 E272*(stop) mutation;dHsaMDS544377207 for the TP53c.994C-1G mutation.

The ddPCR assay conditions for each customised or validated mutation detection assay are listed in Table 2.

### 2.6. Droplet Digital PCR (ddPCR)

A ddPCR System (BioRad Laboratories, Hercules, CA, USA ^TM^) was used for mutational analysis in ctDNA from plasma. After PCR, the signal intensities were read and analysed using Quantasoft™ Analysis Pro Software 1.0.596 (BioRad Laboratories, Hercules, CA, USA ^TM^). Wells that produced less than 10,000 droplets were excluded from the analysis. The goal was to determine the proportion of molecules, with and without a mutation, and to calculate the MAF that equals fractional abundance. The calculations have been described previously [18].

### 2.7. Statistical Analysis

Pearson correlation analysis was used to assess correlations between different biomarkers. To calculate these correlations, measurements of the biomarkers should have been taken on the same dates, but this was not always possible. To overcome this limitation, we obtained approximations of cfDNA and ctDNA values on the dates when CA125 was measured based on linear interpolation of the observed values. Linear interpolation connects two existing observations with a straight line; then, that line is used to estimate the values that the marker would have at other times. While interpolation may not be the best way to determine what is happening between the two observations for which data are available, it is the simplest model.

In addition to individual patient correlation analyses, data from different patients were aggregated to obtain a repeat measurement correlation coefficient. This coefficient considers variability associated within aggregated datasets and tests if there is a common within-individual correlation pattern for paired measures assessed over time for multiple individuals. It is said to test if there is a common correlation pattern between the variables considered over time for all the patients. We used the R package ‘rmcorr’ [20] for this analysis. For all tests, a *p*-value < 0.05 was considered statistically significant.

## 3. Results

### 3.1. Overview

We included 22 patients in the study and sequenced samples of their tumour tissues. Among them, six patients with HGSOC and pathological mutations and who were actively receiving therapy were enrolled to participate in the ddPCR assay. Samples from these six patients with pathological mutations were taken between 8 and 33 months during the course of treatment (Table 1). cfDNA was isolated from a total of 55 samples representing different time points. Based on diagnostic tumour tissue sequencing using a unique three-gene panel, one to two cancer mutations per each patient’s tumour were targeted with customised ddPCR primers. In each assay, the patient’s tumour tissue DNA was used as a positive control for the ddPCR assay (Table 1). Unfortunately, two of the six patients did not have complete records, so they were excluded from the analysis. The four patients with complete records are patients labelled as 5, 8, 10, and 11 in Table 1.

The advantages of ddPCR over traditional qPCR in the context of ovarian cancer is the possibility to quantify ctDNA mutations in plasma samples. Unlike qPCR, which provides relative quantification, ddPCR offers absolute quantification, enabling precise determination of the total number of ctDNA molecules carrying a specific mutation. This allows for a more detailed analysis of ctDNA dynamics and its potential role in disease monitoring and treatment response assessment. The study findings underscore the effectiveness of ddPCR in detecting and tracking ctDNA carrying mutations. We observed changes in ctDNA mutation levels over time in 83% of the patients. This demonstrates the ability of ddPCR to capture subtle fluctuations in ctDNA levels, potentially providing valuable insights into disease progression and treatment efficacy. We compared the ctDNA mutation levels, measured as copies/mL of plasma, to the serum biomarker levels. The number of measurements was different for each case. The cfDNA and ctDNA measurements were taken on the same day, but the CA125 measurements did not necessarily coincide on the same dates, which is why we performed linear interpolation. 

### 3.2. ctDNA Mutations and the Clinical Outcome

Looking at the clinical and analytical evolutions of the four patients with complete records, we detected two patients (patients 5 and 11) with clear disease progression on treatment, one patient (patient 10) with stable disease during follow-up, and a fourth patient (patient 8) with evidence of relapse only in cfDNA, indicating changes in tumour cell subpopulations. Let us analyse the cases separately.

(A)First patient with complete records. (Patient with clear disease progression on treatment)

Figure 1 shows the progression of the disease for patient 5 (according to the numbering shown in Table 1), specifically the treatment course and CT images (Figure 1A), the cfDNA (Figure 1B), the *TP53* R175H (c.524 G>A) mutation (Figure 1C), and the CA125 (Figure 1D) levels. The patient received different treatments—carboplatin, an alkylating agent; carboplatin combined with olaparib, a poly (ADP-ribose) polymerase (PARP) inhibitor; and olaparib for maintenance—consecutively. Over the course of treatment, we quantified the *TP53* R175H (c.524 G>A) mutation. On the day of cytoreduction surgery, there were eight mutated copies/mL of plasma (Figure 1C), and CA125 started to surpass the normal baseline range of 35 U/mL (Figure 1D). On day 196 after the surgery, the number of mutated copies dropped to zero, and CA125 was within the normal range. By day 318, there were three (standard deviation [SD] 2) mutated copies/mL of plasma, and CA125 started to exceed its normal values. On day 400, there were 12 (SD 1) mutated copies/mL, and CA125 exceeded the normal values. Based on CT imaging, the longitudinal evaluation of the cfDNA, ctDNA, and CA125 serum levels, this patient showed disease progression. 

Figure 2 shows the quantification by ddPCR of the number of *TP53* (R175H) copies to assess the response to the treatment established in the patient according to the UR-COAP protocols. Table 3 shows the correlations among the cfDNA, CA125, and *TP53* (R175H) ctDNA for this patient.

(B)Second patient with complete records. (Patient with clear disease progression on treatment)

Patient 11 (according to the numbering shown in Table 1), had a *BRCA1* E272* (c.814 G>T) mutation, in addition to another *TP53* pathogenic mutation—*TP53* R248Q (c.743 G>A)—that had been detected at diagnosis in the tumour tissue (Table 1). This patient was treated with a cisplatin doublet regimen in the neoadjuvant (before surgery) and adjuvant (after surgery) settings. Olaparib maintenance was administered until the end of the study (Figure 3A). CT imaging and CA125 serum levels showed evidence of disease progression on day 625 (Figure 3A,D). Quantification of the *BRCA1* E272* (c.814 G>T) mutation 125 days prior to debulking surgery showed 73 (SD 0) mutated copies/mL of plasma and high CA125 levels (>1000 U/mL) relative to the threshold value (Figure 3B). On the day of the cytoreduction surgery, the number of mutated copies dropped to one (SD 1) copy/mL of plasma, and the CA125 level was still above the threshold value. After an increase to two mutated copies/mL of plasma due to the surgical trauma-induced effect (flare-up effect), mutated copies could not be detected from day 51 to day 352. This could mean that the treatment was effective and that the cytoreduction was probably complete (zero residual tumour). Moreover, the patient’s CA125 levels were within the normal range. It is not until day 687 that mutated copies were measured again (four [SD 3] mutated copies/mL of plasma), and the CA125 serum level slightly exceeded the normal limit (Figure 3C). Based on CT imaging and CA125 levels (RECIST), this patient was diagnosed with disease progression on day 632 after debulking surgery. The data obtained from longitudinal monitoring of ctDNA corroborated this recurrence.

In this patient, the number of *BRCA1* E272* (c.814 G>T) copies was quantified by ddPCR to assess the response to the treatment established according to the UR-COAP protocols. There was a decrease from 72 mutated copies/mL of plasma to 1 mutated copy/mL on the day of surgery. It should be noted that there were no mutated copies 1 month after the surgery, which could objectively show that cytoreduction was complete. Likewise, there was stability up to 12 months after surgery. An increase occurred a few months before the recurrence (24 months; Figure 4). Table 3 shows the correlations among cfDNA, CA125, *TP53* R248Q (c.743 G>A) ctDNA, and *BRCA1* E272* (c.814 G>T) ctDNA.

(C)Third patient with complete records. (Patient with stable disease monitoring)

At the beginning of the diagnostic process, patient 10 underwent laparoscopy for biopsy and to determine the PCI. Likewise, a CT scan was performed for evaluation and to determine the radiological PCI. Both quantifications showed a PCI > 20; therefore, the patient was referred to three cycles of neoadjuvant chemotherapy. CA125 levels and the *TP53* R175H mutation were monitored by ctDNA quantification (Figure 5). At the beginning of neoadjuvant chemotherapy, the CA125 serum level was 1828 U/mL, cfDNA was 12 ng/mL, and there were five mutated copies/mL of plasma of the *TP53* R175H mutation. On day 0 (R0 cytoreductive surgery), there was a decrease in all markers, especially CA125. Based on CT imaging, the PCI was eight. After surgery, the patient completed adjuvant chemotherapy with three more cycles of carboplatin and subsequently began maintenance with PARP inhibitors. After the flare-up effect had subsided, the cfDNA plasma levels and CA125 serum levels remained stable at the subsequent control measurements, but there was a slight increase in the number of *TP53* R175H copies on day 321 that was not confirmed later and was attributed to technical difficulties. The patient is currently asymptomatic and has not shown recurrence. The patient showed significant correlations among CA125, cfDNA, and TP53 ctDNA (Table 3).

(D)Fourth patient with complete records. (Patient with evidence of relapse only in cfDNA, indicating changes in tumour cell subpopulations)

Patient 8 was 60 years old and diagnosed with HGSOC. During the diagnosis, a CT scan was performed to determine the PCI (the value was 16). The initial CA125 serum level was 6418 U/mL. The *TP53* (c.994C-1G) mutation was followed by ctDNA quantification. Initially, the patient had 246 (SD 70) mutated copies/mL of plasma (Figure 6). The International Federation of Gynaecology and Obstetrics indicated primary cytoreduction surgery. Complete surgery was achieved (R0). The FIGO stage was IVb due to a resected hepatic metastasis. Six cycles of adjuvant chemotherapy were administered (carboplatin + paclitaxel), and maintenance was provided with bevacizumab over the course of 15 months, according to the protocol. After 175 days, for *TP53* (c.994C-1G), there was only one mutated copy/mL of plasma and CA125 serum levels were below the normal limit. On day 200, cfDNA began to increase until day 400; however, the number of *TP53* (c.994C-1G)-mutated copies remained at zero. A recurrence was diagnosed on day 625 based on CT imaging. At that time, the cfDNA plasma levels, CA125 serum levels, and number of *TP53* (c.994C-1G)-mutated copies were below the normal range. We speculate that the recurrence was caused by a change in subcellular mutated clones, and the increase in cfDNA occurred 400 days before the recurrence was diagnosed (Figure 12). This was confirmed by the biopsy obtained from the liver recurrence that demonstrated a new mutation of the TP53 gene. The patient was treated with thermoablation for her liver recurrence. There were significant correlations among CA125, cfDNA, and *TP53* (c.994C-1G) ctDNA (Table 3).

### 3.3. Correlations between Individualised ctDNA Mutations and Serum Biomarkers

We determined the CA125 serum levels (U/mL) for the samples taken over time from the 22 patients with HGSOC. For the four patients with complete records, these blood samples were also used for ctDNA analysis. For each patient, we analysed the correlation between the measurements taken over time of cfDNA, *BRCA* mutations, *TP53* mutations, and CA125 serum levels. Moreover, for each patient, we calculated the Pearson correlation coefficient between the mean number of mutated ctDNA copies and the CA125 serum levels over time (Table 3). The CA125 levels were 7.2–11,100 U/mL, and for ctDNA, there were 0–246 mutated copies/mL of plasma.

All four patients show significant positive correlations between CA125 and cfDNA over time (in all cases *p* ≤ 0.04) and also between CA125 and TP53ctDNA (in all cases *p* ≤ 0.02). Therefore, for each patient individually, there is a direct relationship between her cfDNA and CA125 levels over the follow-up, and also between her ctDNA and CA125 levels.

By pooling the data from the four patients, we can test if there is a common pattern of correlation over time for all of them. The result of the repeated measures correlation analysis (rmcorr) is shown in Figure 7. This tests if there is a common within-individual association for paired measures assessed over time for multiple individuals. 

The *p*-value corresponding to the aggregated correlation between cfDNA and CA125 is 0.14 (Figure 7A). This indicates that our data cannot detect a common correlation pattern between cfDNA and CA125 for the four patients. However, the *p*-value corresponding to the aggregated correlation between TP53ctDNA and CA125 is 0.01 (Figure 7B), indicating a common correlation pattern between TP53ctDNA and CA125 for the four patients.

The same can be observed when we include more patients in the analysis, even if they do not have complete records. Thus, Figure 8A,B show the *p*-values corresponding to the aggregated correlations between cfDNA and CA125 and between TP53ctDNA and CA125 for eleven patients of the dataset (the four patients with complete records and another seven patients with partial records). Once again, a common correlation pattern between TP53ctDNA and CA125 is detected for the 11 patients (Figure 8B, *p*-value = 0.03), although the *p*-value for the aggregated correlations between cfDNA and CA125 is non-significant (Figure 8A, *p*-value = 0.23), indicating that there may be different correlation patterns between cfDNA and CA125 for different groups of patients. 

In our opinion, the fact that there is no common correlation pattern between cfDNA and CA125 for all patients opens up an interesting research question. Do different correlation patterns exist between cfDNA and CA125, depending on the evolution of the disease during follow-up? 

As stated above, among the four patients with complete records, we detected two patients (patients 5 and 11) with clear disease progression on treatment, one patient (patient 10) with stable disease during follow-up, and a fourth patient (patient 9) with changes in tumour cell subpopulations. Figure 9A,B show the aggregated correlations for the measurements of the two patients with clear disease progression on treatment. We obtained a positive and significant aggregated correlation between CA125 and cfDNA (r = 0.88; *p* = 0; Figure 9A) and a positive non-significant correlation, but very close to the statistical significance between CA125 and *TP53* ctDNA (r = 0.65; *p* = 0.058; Figure 9B). Therefore, a larger dataset would be needed to group patients into categories according to their clinical and analytical evolutions and to analyse the correlation between ca125 and cfDNA in these categories.

To determine whether cfDNA detects recurrence earlier than the conventional CA125 biomarker, we examined both biomarkers over time in patients 5 and 11, i.e., those patients with clear disease progression on treatment. A frequently used approach to identify shifts and changes in trends within time series data involves analysing differences. In other words, the data are analysed by subtracting the value of each observation (observed in time t) from that of the preceding one (observed in time t − 1). Figure 10 shows the differenced series of cfDNA and CA125 for patient 5. The small negative values of the differenced series indicate that some observations decreased slightly over time (observation at time t < observation at t − 1), and the positive values indicate that some observations increased over time (observation at time t > observation at t − 1). The greater the value of the difference, the higher the growth rate. In patient 5, cfDNA started to increase on day 196 and stayed elevated until day 318. CA125 started to increase on day 136, but the strongest growth occurred after day 321. Thus, cfDNA provided evidence of a possible recurrence more than 100 days before the recurrence was confirmed by CA125 (Figure 10). In patient 11, cfDNA decreased and showed stable behaviour until around day 350, long before the diagnosed recurrence and long before CA125 elevation indicated signs of recurrence. Indeed, CA125 began to increase on day 528, and recurrence was diagnosed on day 632 (Figure 11).

Patient 8 shows evidence of relapse only in cfDNA, indicating changes in tumour cell subpopulations. Figure 12 shows that cfDNA started increasing at 175 days; however, CA125 began increasing on day 532, but the most pronounced growth occurred after day 630. The increase in cfDNA occurred 400 days before the recurrence was diagnosed (Figure 12).

## 4. Discussion 

cfDNA showed an important correlation with tumour recurrence. Furthermore, cfDNA anticipated the recurrence more than 6 months before it was diagnosed based on RECIST. Currently, there is no objective method available to determine whether the entire visible tumour has been removed. In all cases with complete cytoreduction (R0), the amount of cfDNA after surgery decreased with respect to the amount before surgery, once the post-surgery flare-up effect was overcome. However, the only objective way to define complete cytoreduction is determined to be when there are no mutated ctDNA copies based on ddPCR. 

Ovarian cancer has a high mortality rate of 64.4% [19]. About 80% of patients will relapse despite chemotherapy treatment (CHT) and targeted maintenance therapy [20]. 

To date, there is no established biomarker capable to predict response to chemotherapy or diagnose a recurrence at an early stage. cfDNA showed an important correlation with tumour recurrence. Furthermore, cfDNA anticipated the recurrence more than 6 months before it was diagnosed based on RECIST, even increasing prior to elevation of the CA125 marker [11].

Similar to other authors, we believe that the determination of genetic mutations that predominate in ovarian cancer can be used as a target for treatment because drug resistance to chemotherapy occurs during progression of the disease [21]. We found that dynamic variations in ctDNA levels reflect changes in the tumour burden and can predict disease progression several months before it is apparent on imaging or with biochemical tests, similarly to a previous study [11]. Based on our results, the first and the second patients would benefit from early diagnosis of recurrence before it is clinically detected. In this category, cfDNA increases early, likely because it is very sensitive to changes in free DNA, but the detection of mutated ctDNA will accurately confirm the recurrence. 

However, ovarian cancer has a high heterogeneity in the mutations presented, and even the different histological types have different genetic characteristics, which could, in the future, lead to advances in treatment [22].

In all phases of cancer management, it is necessary to monitor treatment response to prevent continuing therapies that lack efficacy and to avoid unnecessary side effects. In this context, molecular analysis is used to select appropriate therapies based on a patient’s cancer genome. The treatment choice for recurrent disease is based on the molecular alterations of the primary tumour. Therefore, understanding these molecular alterations in tumour recurrence by using a minimally invasive technique compared with a biopsy of the metastatic lesion is a promising technique to personalise care for patients with ovarian cancer.

Resistance to chemotherapy treatment is one of the main causes of relapse and death in these patients. Cellular instability allows rapid adaptation to the local molecular microenvironment using innate and acquired resistance mechanisms. These cells are capable of reversing mutations such as the homologous repair pathway or secondary somatic mutations in BRCA 1/2 leading to platinum resistance and iPARPs [23].

cfDNA can vary according to the response to chemotherapy and relate to tumour recurrence [24]. However, the proportion of cfDNA originating from tumour cells is not just determined by the state and size of the tumour [25]; clearance, DNA degradation, lymphatic circulation, and blood processing also contribute [21]. Quantification of mutated ctDNA copies in the plasma of patients with ovarian cancer could also be useful during follow-up because it could improve specificity. In another study, researchers assessed breast tumour-specific mutations in 55 patients who had undergone surgery and chemotherapy as a curative treatment. The results showed patients at risk of recurrence may be identified earlier. The presence of ctDNA predicted recurrence in 12 out of the 15 patients who relapsed. Among patients who did not relapse, 96% had no measurable ctDNA in the post-surgery sample [26]. These findings are similar to the third patient we described in this paper. In addition, several studies have shown that ctDNA analysis could be used to monitor the emergence of multiple resistant clones during the treatment course [27,28,29,30,31,32].

Otsuka et al. [33] also studied patients with ovarian cancer. They identified *TP53* mutations, and in 2 of 12 cases, they detected identical mutations in the DNA of their preoperative plasma. This genomic alteration was undetected after surgery and follow-up of these two patients. However, in one case, the *TP53* mutation reappeared 16 months after surgery, indicating that the detection of mutant cfDNA might be promising for monitoring of the treatment efficacy. This eventuality is consistent with the fourth patient in our study. The patient presented recurrence due to a change or the appearance of a new mutation in the tumour not detected by ddPCR, but the elevation of cfDNA does warn of possible recurrence.

In an example of the variability in the molecular profile in ovarian cancer, Forshew et al. [34] re-sequenced tumour tissue from a right oophorectomy specimen from a patient with ovarian cancer at the time of primary cytoreduction surgery. The authors identified a *TP53* mutation, and they evaluated ctDNA levels. As the cancer progressed, ctDNA analysis showed the appearance of an epidermal growth factor (EGFR) mutation in the plasma samples. This mutation was not detected in the original oophorectomy tissue sample.

The molecular heterogeneity within advanced ovarian cancer is demonstrated in the case of extensive intraperitoneal dissemination. Molecular profiling of primary tumours and metastases may be useful in identifying the biology underlying ovarian cancer progression [35].

Some authors have demonstrated that somatic genetic alterations can be detected by a massive parallel sequencing-based analysis of ctDNA from the plasma of patients with cancer. Many of these genetic alterations are related to tumour prognosis. Genetic alterations have been described, such as the MECOM gene related to cell development and differentiation, more frequent in leukaemia, which when present in ovarian cancer, is associated with a better prognosis and a longer disease-free interval. Similarly, a high expression of the MELK gene is associated with better overall survival [23]. 

Unfortunately, it is still too expensive to apply as part of routine clinical practice [32,34,36]. In another study, the authors found that the gene associated with molecular processes of epithelial to mesenchymal transition may play a significant role in cisplatin resistance in ovarian cancer [37].

Relating this mesenchymal differentiation is a necessary process for the development of metastasis, with this subtype being associated with shorter survival [38].

To highlight the identification of genes related to the immune response, the expression of CXCL9, a chemokine that mediates T-cell recruitment, is a marker of favourable prognosis, with greater survival being related to greater lymphocyte infiltration, which could allow the development of another therapeutic option [23].

The fact that a plasma analysis of the mutations in cfDNA could identify heterogeneous clones strengthens the hypothesis that chemotherapy, like PARP inhibitors, could select for resistant clones that were initially present only at a low frequency or perhaps were not even present in the primary tumour. DNA quantification provides guidance towards the diagnosis of tumour recurrence and sequencing together with quantification that will allow us to carry out targeted treatments. Screening for the presence of genomic abnormalities in ctDNA is more practical than metastatic tumour biopsies, is suitable for all patients, and is more economical and more comfortable for patients. Finally, screening ctDNA provides a current assessment of the genetic profile of recurrent cancer, allowing the identification of genetic events selected by prior treatments [11,29,31].

To avoid continuing inefficacious therapies and to prevent side effects, it is necessary to detect the appearance of resistance to different systemic treatments and targeted agents leading to the recurrence of ovarian cancer. In our sample, patient 11 showed a recurrence that did not present the *BRCA1* mutation that was present at diagnosis, possibly due to maintenance treatment with a PARP inhibitor, with an increase in cfDNA, possibly at the expense of the tumour clone with the *TP53* mutation. Similarly, patient 8 showed an increase in cfDNA without an increase in the *TP53* mutation detected at the start of the disease.

cfDNA represents a potential surrogate for the tumour based on non-invasive serial monitoring of the tumour genome [39]. ddPCR and NGS of liquid biopsies may be feasible to promote earlier detection of recurrences, ensuring treatment decisions and optimal management in ovarian cancer. However, increased levels of ctDNA may be found in patients with benign lesions, inflammatory diseases, and tissue trauma. Hence, it is necessary to identify the levels related to malignant disease [40,41]. Moreover, a ctDNA-based analysis for ovarian cancer is currently limited to hotspot mutations, structural rearrangements, copy number alterations, and changes in DNA methylation. These may not be sufficient for the therapeutic management of ovarian cancer tumours.

The strengths of the present study include the homogeneous cohort of patients and the single uniform approach to care for patients with advanced ovarian cancer. It should be noted that the same patient has both a cfDNA and ctDNA analysis that allows us to detect the recurrence and the tumour clone responsible for it. In this way, we can demonstrate that our findings could be translated to the clinic. The findings presented in this article open up new fields of research, suggesting the use of this technique for the early diagnosis of ovarian cancer, where transvaginal ultrasound and the CA125 marker have not demonstrated the correct sensitivity and specificity to be used for population screening. 

However, this study has some limitations. Indeed, the population evaluated is small, 22 patients, with a short follow-up, only 48 months, so we do not have any deaths, and the study only allowed us to identify a small proportion of women who relapsed during the study (6 of 22 patients).

Sixty percent of the patients analysed had received neoadjuvant chemotherapy, which means that these patients have a worse prognosis and a higher risk of recurrence, so we could over-estimate the reliability of the test. 

In addition, we did not compare with the HE4 marker (human epididymal protein), a marker that has been established in clinical practice in the diagnosis and follow-up of patients with ovarian cancer. 

Finally, one of the limitations that opens up new fields of study is whether we should treat these patients with cfDNA or ctDNA elevation or wait for evidence of recurrence by imaging. 

Instead of these limitations, the benefit of using cfDNA and ctDNA and evaluating the genomic profile of the tumour can help personalise treatment for patients with ovarian cancer.

## 5. Conclusions

The constant development and discovery of new molecular biomarkers is a promising method to treat patients with ovarian cancer. An analysis of cfDNA and ctDNA offers a non-invasive clinical tool for real-time monitoring of primary tumours and the metastatic disease burden. Using liquid biopsies to quantify cfDNA and ctDNA could indicate recurrence several months earlier than traditional clinical methods and also provide an objective definition of complete cytoreduction. The use of liquid biopsies could also allow choosing the therapy according to the genetic changes correlated with the therapeutic response. 

## Figures and Tables

**Figure 1 diagnostics-14-01868-f001:**
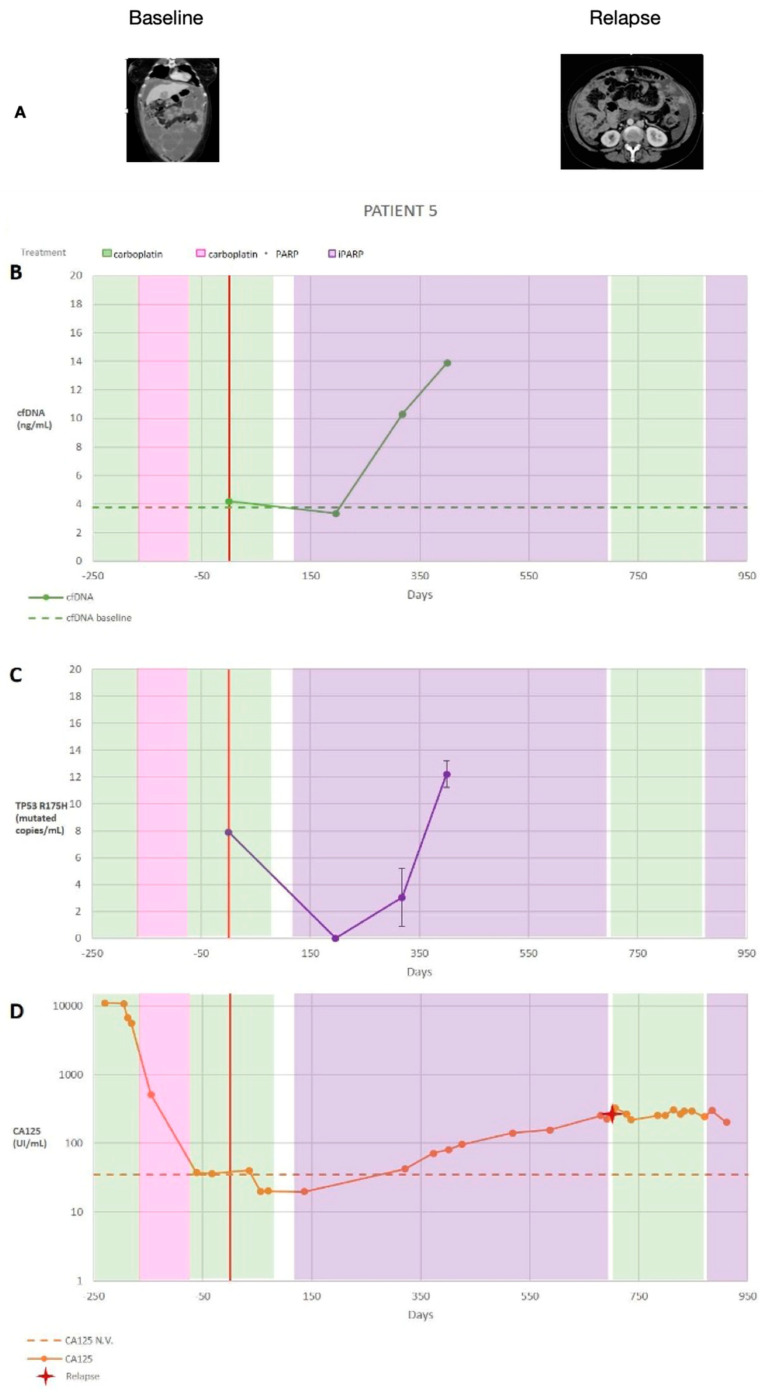
(**A**) Summary of patient 5’s clinical information, including the treatment type and dates (treatment dates indicated by the coloured vertical stripes; the date of the cytoreduction surgery is indicated by a red vertical line, and the date of the diagnosed recurrence is indicated by a red cross) as well as computed tomography images and summarised results. (**B**) Cell-free DNA (cfDNA) plasma levels are presented as ng/mL. (**C**) Circulating tumour DNA (ctDNA) plasma levels are expressed as the number of mutated copies/mL of plasma. Replicates were used to quantify the average *TP53* R175H copies for each time point, and the error bars represent the standard deviation across replicates; no bar indicates that the standard deviation was too low to be visualised on the scale used. The volume-adjusted limit of detection was 21.2 GEs (genome equivalent)/mL for all time points. A two-tailed *t*-test was used to compare the ctDNA levels between sequential time points. (**D**) Cancer antigen 125 (CA125) serum levels (U/mL); data from multiple replicates were not provided. * *p* = 0.05.

**Figure 2 diagnostics-14-01868-f002:**
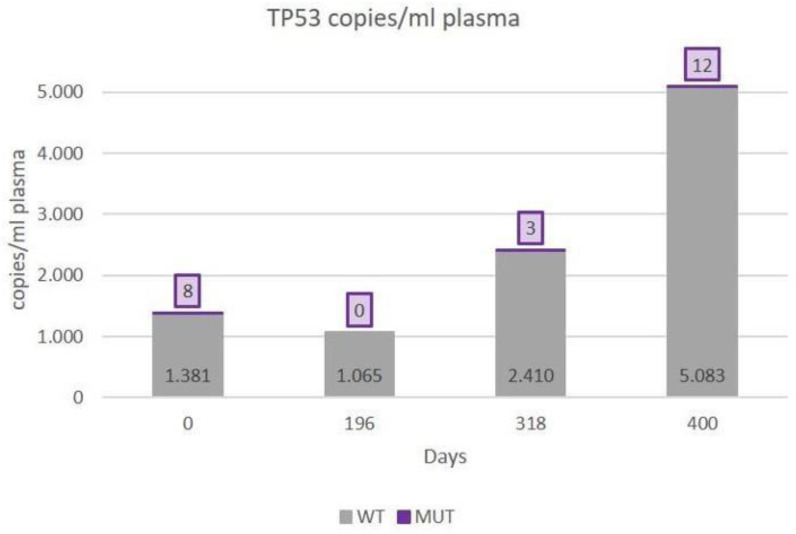
The number of *TP53* (R175H) copies quantified by digital droplet polymerase chain reaction.

**Figure 3 diagnostics-14-01868-f003:**
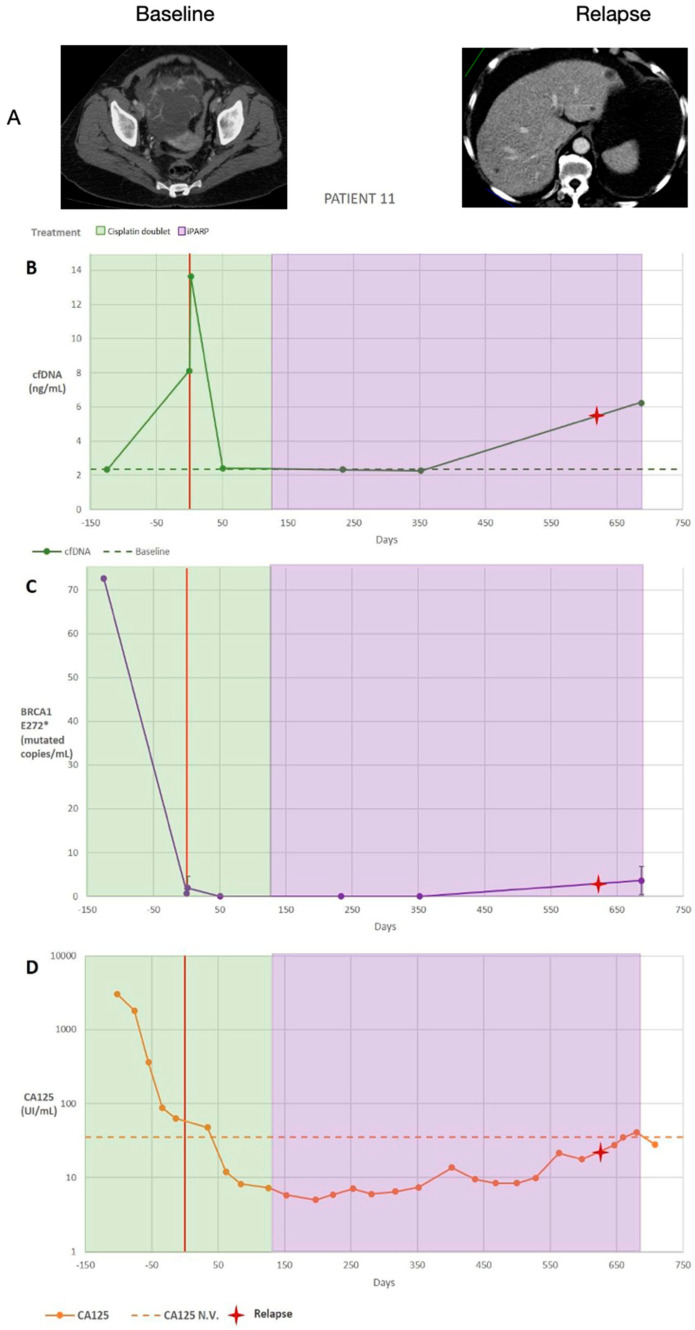
(**A**) Summary of patient 11’s clinical information, including the treatment type and dates (treatment dates indicated by the coloured vertical stripes; the date of the cytoreduction surgery is indicated by a red vertical line, and the date of a diagnosed relapse is indicated by a red cross) as well as computed tomography images and summarised results. (**B**) Cell-free DNA (cfDNA) plasma levels are presented as ng/mL. (**C**) Circulating tumour DNA (ctDNA) plasma levels are expressed as the number of mutated copies/mL. Replicates were used to quantify the average *BRCA1* E272* (c.814 G>T) levels for each time point, and the error bars represent the standard deviation across replicates; no bars indicate that the standard deviation was too low to be visualised on the scale used. The volume-adjusted limit of detection was 21.2 GE/mL for all time points. A two-tailed *t*-test was used to compare the ctDNA quantities between sequential time points. (**D**) Cancer antigen 125 (CA125) serum levels (U/mL); data from multiple replicates were not provided. * *p* < 0.05, ** *p* < 0.01, and *** *p* < 0.001.

**Figure 4 diagnostics-14-01868-f004:**
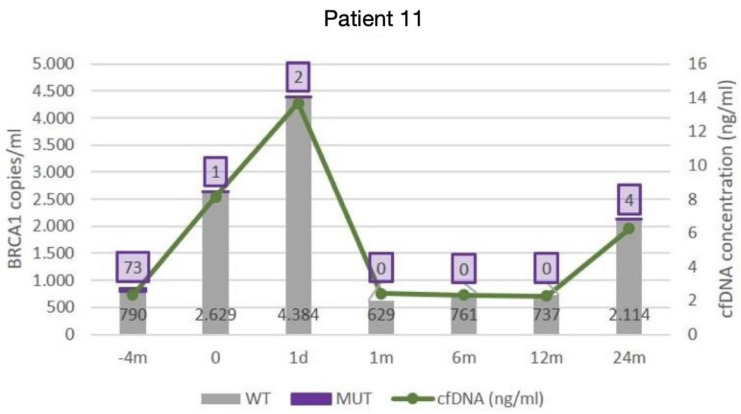
The number of *BCRA1* E272* (c.814 G>T) copies quantified by digital droplet polymerase chain reaction.

**Figure 5 diagnostics-14-01868-f005:**
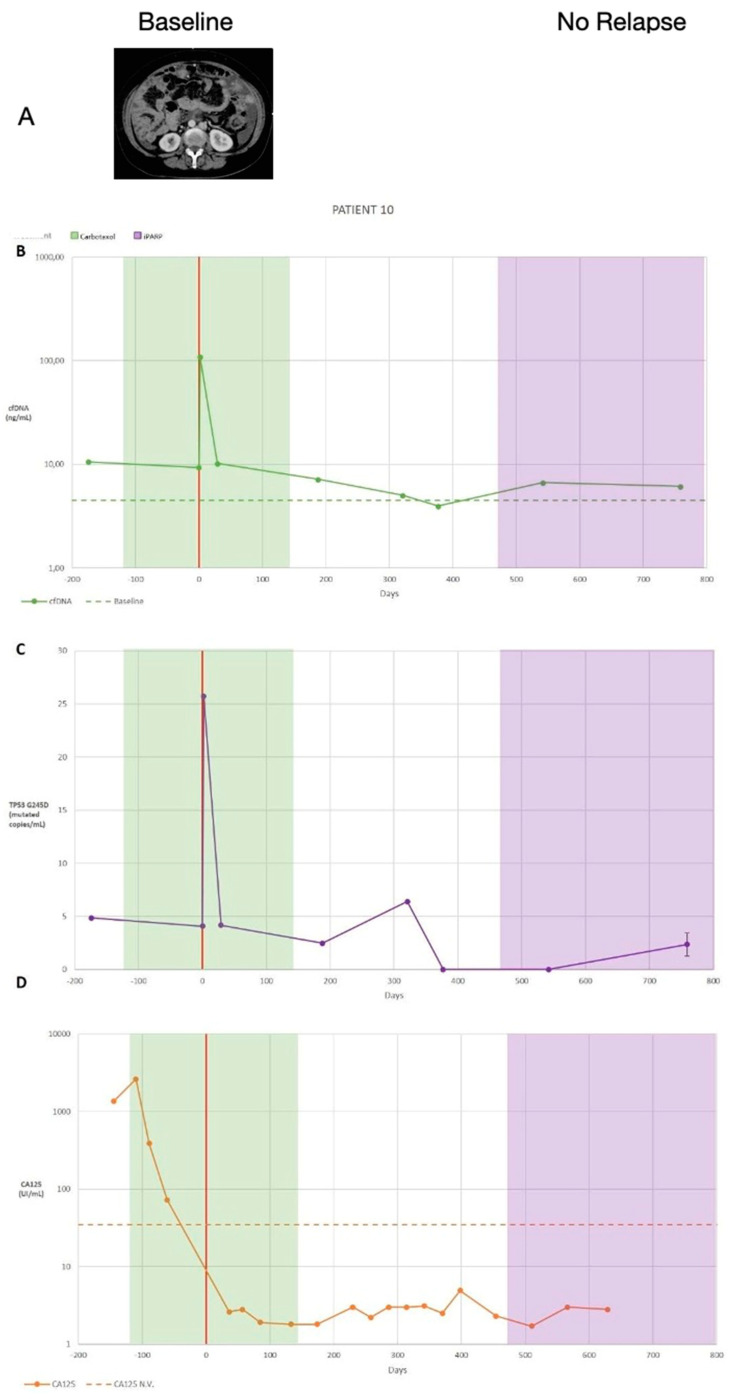
(**A**) Summary of patient 10’s clinical information, including the treatment type and dates (treatment dates indicated by the coloured vertical stripes, and date of the cytoreduction surgery indicated by a red vertical line) as well as a computed tomography scan and summarised results. (**B**) Cell-free DNA (cfDNA) plasma levels are presented as ng/mL. (**C**) Circulating tumour DNA (ctDNA) plasma levels are expressed as the number of mutated copies/mL. Replicates were used to quantify the average TP53 G245 levels for each time point, and the error bars represent the standard deviation across replicates (*n* = 3); no bars indicate that the standard deviation was too low to be visualised on the scale used. The volume-adjusted limit of detection was 21.2 GE/mL for all time points. A two-tailed *t*-test was used to compare the ctDNA quantities between sequential time points. (**D**) Cancer antigen 125 (CA125) serum levels (U/mL); data from multiple replicates were not provided. * *p* < 0.05, ** *p* < 0.01, and *** *p* < 0.001.

**Figure 6 diagnostics-14-01868-f006:**
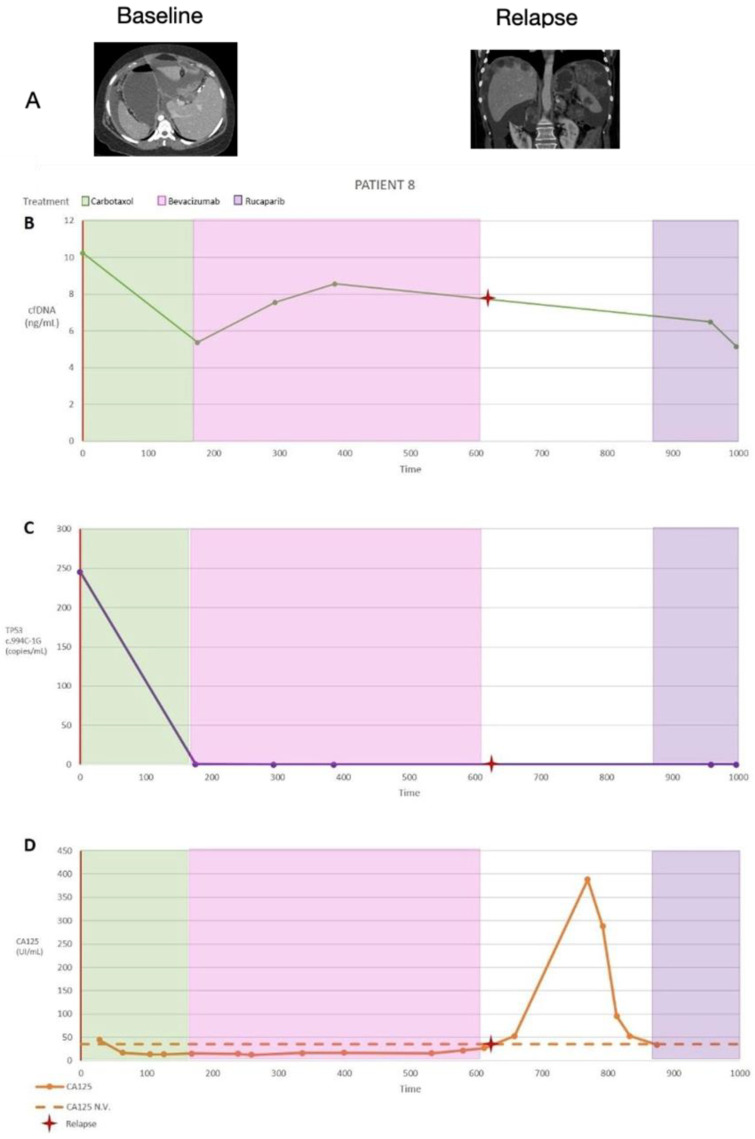
(**A**) Summary of patient 8’s clinical information, including the treatment type and dates (treatment dates indicated by the coloured vertical stripes; the date of the cytoreduction surgery is indicated by a red vertical line, and the date of a diagnosed relapse is indicated by a red cross) as well as computed tomographic images and summarised results. (**B**) Cell-free DNA (cfDNA) plasma levels are presented as ng/mL. (**C**) Circulating tumour DNA (ctDNA) plasma levels are expressed as the number of mutated copies/mL. Replicates were used to quantify the average *TP53* (c.994C-1G) levels for each time point, and the error bars represent the standard deviation across replicates (n = 3); no bars indicate that the standard deviation was too low to be visualised on the scale used. The volume-adjusted limit of detection was 21.2 GE/mL for all time points. A two-tailed *t*-test was used to compare the ctDNA quantities between sequential time points. (**D**) Cancer antigen 125 (CA125) serum levels (U/mL); data from multiple replicates were not provided. * *p* < 0.05, ** *p* < 0.01, and *** *p* < 0.001.

**Figure 7 diagnostics-14-01868-f007:**
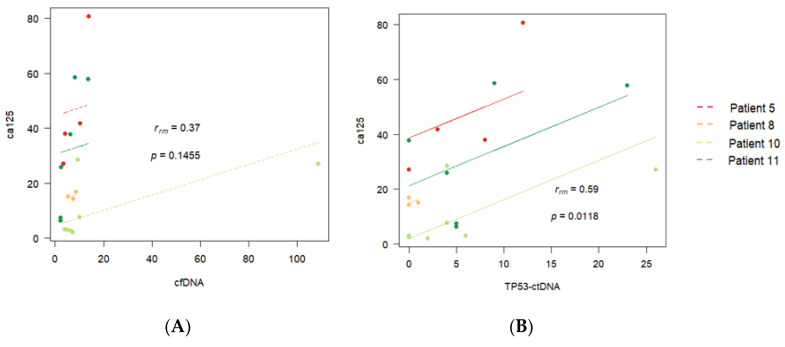
Aggregated correlation between (**A**) cell-free DNA (cfDNA) and CA125 and (**B**) *TP53-*ctDNA and CA125 for the four patients with complete records.

**Figure 8 diagnostics-14-01868-f008:**
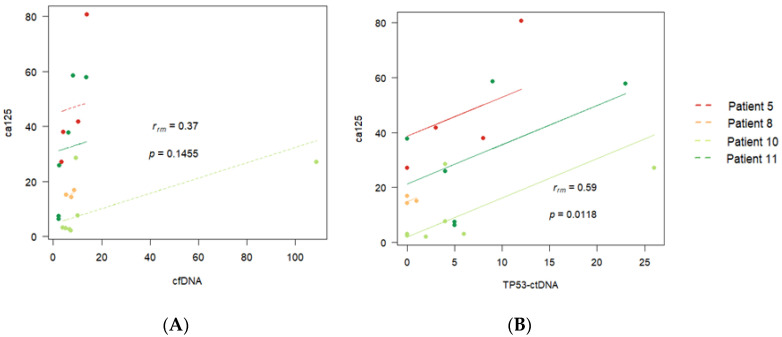
Aggregated correlation between (**A**) cfDNA and CA125 and (**B**) *TP53*-ctDNA and CA125 for eleven different patients of the dataset.

**Figure 9 diagnostics-14-01868-f009:**
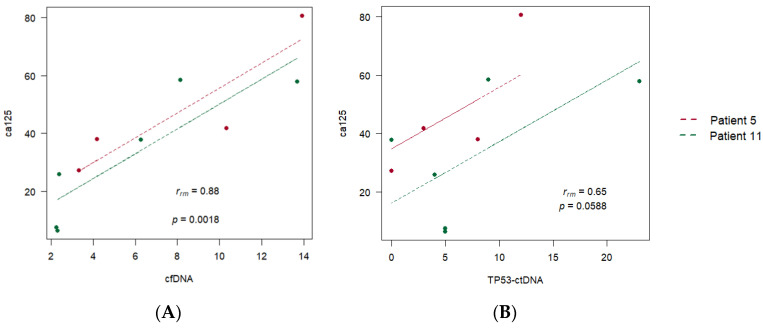
Aggregated correlation between (**A**) cfDNA and CA125 and (**B**) *TP53* ctDNA and CA125 for two patients with the same disease progression on treatment.

**Figure 10 diagnostics-14-01868-f010:**
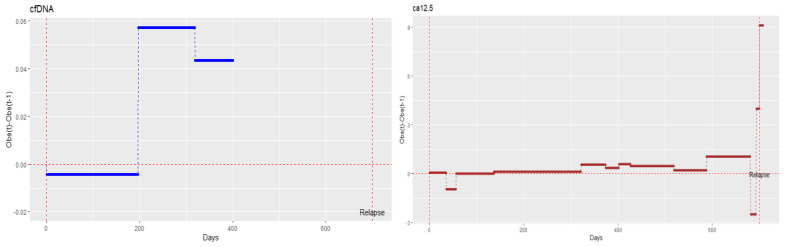
Monitoring cell-free DNA (cfDNA) and cancer antigen 125 (CA125) over time in patient 5.

**Figure 11 diagnostics-14-01868-f011:**
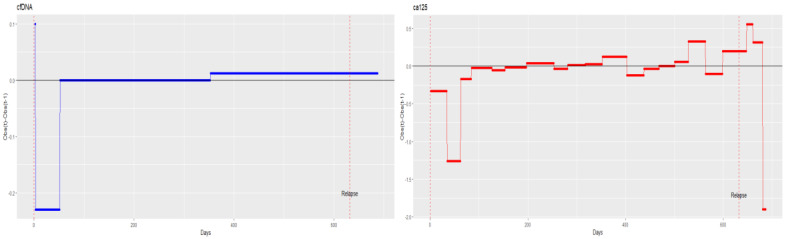
Monitoring cell-free DNA (cfDNA) and cancer antigen 125 (CA125) over time in patient 11.

**Figure 12 diagnostics-14-01868-f012:**
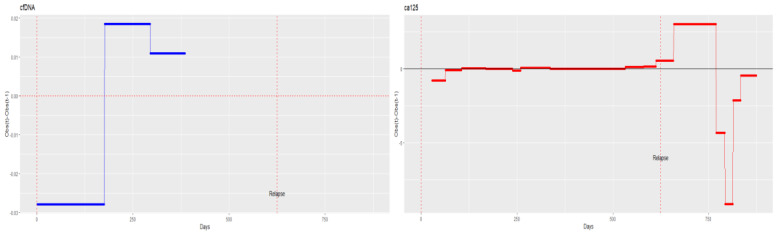
Monitoring cell-free DNA (cfDNA) and cancer antigen 125 (CA125) over time in patient 8.

**Table 1 diagnostics-14-01868-t001:** Clinical cohort overview.

PATIENT	Age	Sample Time Points	Study Length (Days)	Neo-adyuvance	FIGO	PCI	R Surgery	Relapse	Origin	Exitus	Mutation(s) Identified by NGS	Mutation-Specific ctDNA Detected by dPCR (Líquid Biopsy)	Verified in Tumor by dPCR (Tumor Tissue)	Further NGS-Identified Pathogenic Mutations	Further NGS-Identified Mutations Classified as VUS (Variant of Uncertain Significance)
**1**	61	5	538	YES	IIIC	30	R1	YES	Bulgarian	YES				TP53 C135Sfs*35	
**2**	42	5	391	YES	IVA	18	R1	YES	Spanish	YES				BRCA2 G2793V	
**3**	61	6	882	NO	IIIC2	2	R0	NO	Spanish	NO					
**4**	72	5	401	YES	IVB	11		YES		YES					TP53 P27L
**5**	70	4	400	YES	IIIA2	7	R0	YES	Spanish	NO	TP53 R175H	+	+	BRCA1 T688*	
**6**	80	4	384	YES	IIB	23	R0	YES	Spanish	NO					
**7**	70	3	301	NO	IVB	9	R1	NO	Spanish	NO	TP53 R175H BRCA1 E1210R	+	+		
**8**	60	6	997	NO	IVB	18	R0	YES	Spanish	NO	TP53 p.?	+	+		
**9**	63	4	357	YES	IIIC	12		YES		YES				TP53 A86Hfs*37 BRCA2 K3326*	
**10**	53	9	933	YES	IIIA2	16	R0	NO	Moroccan	NO	TP53 G245D	o	+	BRCA1 G1770V	
**11**	51	7	812	YES	IIIC	21	R0	YES	Romanian	NO	BRCA1 E272* TP53 R248Q	+ +	+ +		
**12**	75	7	835	YES	NA	35	R0	YES	Spanish	NO					
**13**	75	4	356	NO	IIIC	13	R0	YES	Spanish	NO				TP53 c.911_919+7delCTAAGCGAGGTAAGCA	
**14**	51	4	532	YES	IIIA	22	R0	NO	Chinese	NO	TP53 Y220C	+	+	BRCA2 S611Yfs*5	BRCA1 K654R
**15**	53	4	365	NO	IC	10	R0	NO	Spanish	NO				BRCA1 Y655* C994*	
**16**	74	7	314	NO	IIIA	17	R0	NO	Spanish	NO	TP53 R249S	+	+	TP53 R254S	
**17**	64	5	288	NO	IIIB	7	R0	NO	Spanish	NO					
**18**	59	5	300	NO	IIIc	12	R0	NO	Spanish	NO				TP53 C275F	
**19**	63	5	280	YES	IVa	19	R1	YES	Spanish	NO				BRCA2c.8332-1G>A = p.?	BRCA2 G2270R E2832K S2984L BRCA1 E438K S186F
**20**	72	4	360	NO	IVa	15	R1	NO	Spanish	NO				BRCA1 S1428*	
**21**	70	5	420	YES	IIIc	20	R0	NO	Romanian	NO				TP53 Y236*	BRCA2 D1441N BRCA1 P1010S
**22**	68	5	400	YES	IIIc	18	R0	YES	Moroccan	NO				TP53 H193R	BRCA2 A2851L

FIGO stage: International Federatio of Gynaecology and Obstetrics staging system. PCI: Peritoneal carcinomatosis index. Cytoreductive Surgery: R0 No residual tumour, R1: Residual tumour less than 1 cm. + means Yes; * is only a variation of the gene.

**Table 2 diagnostics-14-01868-t002:** Thermal cycling conditions for the mutation detection assay.

Cycling Step	Temperature (°C)	Time (s)	Number of Cycles
Enzyme activation	95	600	1
Denaturation	94	30	40
Annealing/extension	50–60 * (optimum)	60 **	40
Enzyme deactivation	98	600	1
Hold	4	Infinite	1

* Temperature of 55 °C for validated assays. ** Check/adjust ramp rate setting to −2 °C/s.

**Table 3 diagnostics-14-01868-t003:** Correlations among cfDNA, *TP53* (R175H) ctDNA, and CA125 for patient 5. Correlations among cfDNA, *TP53* R248Q (c.743 G>A) ctDNA, *BRCA1* E272* (c.814 G>T) ctDNA, and CA125 for patient 11. Correlations among cfDNA, *TP53* G245 ctDNA, and CA125 for patient 10. Correlations among cfDNA, *TP53* (c.994C-1G) ctDNA, and CA125 for patient 8.

Patient	Correlation between:				
	*cfDNA-TP53* (R175H)*ctDNA*		*cfDNA-*CA125	*TP53* (R175H) *ctDNA-CA125*	
5	0.53 (*p* = 0.09)		0.88 (*p* < 0.001)	0.73 (*p* = 0.011)	
Patient	Correlation between:				
	*cfDNA-TP53* R248Q (c.743 G>A) ctDNA	*cfDNA- BRCA1* E272* (c.814 G>T) ctDNA	*cfDNA-*CA125	*TP53* R248Q (c.743 G>A) ctDNA-*CA125*	*BRCA1* E272* (c.814 G>T) ctDNA-*CA125*
11	0.58 (*p* = 0.001)	0.59 (*p* = 0.001)	0.87 (*p* < 0.01)	0.43 (*p* = 0.02)	0.52 (*p* = 0.005)
Patient	Correlation between				
	*cfDNA-TP53* G245 ctDNA		*cf*DNA-CA125	*TP53* G245 ctDNA-CA125	
10	0.93 (*p* < 0.001)		0.68 (*p* < 0.001)	0.66 (*p* < 0.001)	
Patient	Correlation between				
	cfDNA-TP53 (c.994C-1G) ctDNA		*cfDNA-*CA125	TP53 (c.994C-1G) ctDNA-CA125	
8	0.59 (*p* = 0.05)		0.61 (*p* = 0.04)	0.71 (*p* = 0.01)	

## Data Availability

The data that support the findings of this study are available from the corresponding author upon reasonable request.

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
