# Peer review of "The Impact of Liquid Biopsy in Advanced Ovarian Cancer Care"

_diagnostics, 2024, doi:10.3390/diagnostics14171868_

Round 1

Reviewer 1 Report

Comments and Suggestions for Authors

The authors describe a method for the detection of cftDNA using qPCR and digital PCR, which they demonstrate in 22 patients with serous high-grade ovarian carcinoma. Analyses of the primary tumor were performed using FFPE material. The authors use this procedure as follow-up monitoring. Blood samples were taken after 1, 6, 9, 12 and 24 months. Finally, cfDNA was determined by qPCR and ctDNA by ddPCR and compared with CA125 level at seven patients. The authors actually recorded changes in ctDNA mutations in 83% of patients during the course of therapy. The reviewer considers the classification of patients into four groups based on the results to be rather risky due to the small number of cases. It is questionable how representative the significant correlations between TP53, BRCA and CA125 actually are. Because this is only shown in less cases.

The units of the CA125 tests are wrong in the introduction. The description of the sampling between eight months before surgery and 33 months afterward does not correspond to the sampling points described previously. The authors should provide scientific evidence why the boundary between germline mutations and somatic mutations is 45% MAF. Authors should name the sequence of the primers or name the commercial primer set used. A weakness of this statistical evaluation is the regression of the DNA data with the CA-125 data. Firstly, CA-125 is only a conditionally meaningful marker for ovarian cancer. Secondly, the r-value is unusably poor compared to TP53. For non-expert readers, a legend to Table 1 should be available explaining the abbreviations (e.g. PCI, or R Surgery). Section A in Figure 3 is missing and has probably slipped into the text on page 9. The title in Fig. 4 cannot be reassigned. The designations A and B are missing in Fig. 5. The positive correlation in Fig. 5 B is to be rejected with an r-value of 0.65. The claim of a correlation is unscientific with these values.

The reviewer does not rule out that the results of this study provide a good hint of indicative markers that may be more effective than CA125 alone. How the authors discuss, this must also be discussed and requires a meaningful study. The authors correctly state that previous successes in detecting tumor-relevant mutations are too expensive. The authors should it make more clear, that detecting specific mutations requires precise knowledge of what the clinician is looking for and a corresponding ddPCR test. This would be very time-consuming, as the authors themselves confirm the high variability of the mutations. The fact that this procedure is unlikely to lead to a broadly applicable detection test is the real statement that should be discussed with these data. To prove that cfDNA can indicate recurrences, as described here, larger studies are needed.  

The reviewer recognizes the importance of this study. However, this manuscript is only acceptable after a major, detailed revision.

Author Response

Prof. Dr. Andreas Kjaer

Editor-in-Chief

Diagnostics

Manuscript ID: diagnostics-3058250

Title: The Impact of Liquid Biopsy in Advanced Ovarian Cancer Care

10 July 2024

Dear Prof. Dr. Andreas Kjaer,

I have received your mail with the reviewer´s reports about our work. Attached please find the reviewers’ lined comments along with my comments explaining why I agree or disagree with their opinions.

Reviewer 1

The authors describe a method for the detection of cftDNA using qPCR and digital PCR, which they demonstrate in 22 patients with serous high-grade ovarian carcinoma.

Analyses of the primary tumor were performed using FFPE material. The authors use this procedure as follow-up monitoring. Blood samples were taken after 1, 6, 9, 12 and 24 months. Finally, cfDNA was determined by qPCR and ctDNA by ddPCR and compared with CA125 level at seven patients.

The authors actually recorded changes in ctDNA mutations in 83% of patients during the course of therapy.

1.- The reviewer considers the classification of patients into four groups based on the results to be rather risky due to the small number of cases.

Answer 1.- There are 3 groups, not 4. We have classified them into 3 groups because we have patients who evolve in 3 different ways and who will mark the future lines of research of this preliminary work.

2.-It is questionable how representative the significant correlations between TP53, BRCA and CA125 actually are. Because this is only shown in less cases.

Answer 2.- Of the 22 cases we are left with 4 that we fear complete follow-up. These correlations are shown in the 4 cases studied and these correlations are significant even though there is little data for each patient. When there is little data, the tests are usually conservative and do not show significant differences, however, in this case they do find them, which shows that the correlation exists.

3.-The units of the CA125 tests are wrong in the introduction.

Answer 3. Thank you very much for your observation, we have corrected the units to milliliters (U/ml)

4.-The description of the sampling between eight months before surgery and 33 months afterward does not correspond to the sampling points described previously.

Answer 4: Methods For each patient, whole blood samples (10 mL) were collected on a fixed schedule: the date of the diagnostic laparoscopy (t-1), the date of debulking surgery (considered the day 0; t0), 24 - 48 hours after the surgery (t1), and one, six, nine, 12 and 24 months after surgery (t2 - t6). The number of total time points sampled for each patient is listed in Table 1. Tumor tissue was collected at t0 or t-1 for diagnostic purposes.

As described in methods, this is how the study was designed, but in the reality of clinical care, the dates of sample collection were adapted to the patient's clinical status (complications, side effects of medication, etc.), which is why we say that we have more samples. beyond 24 months.

5.-The authors should provide scientific evidence why the boundary between germline mutations and somatic mutations is 45% MAF.

Answer 5 Since homozygote germ-line mutations have MAFs of 100% and heterozygote ones an expected MAF of 50% we considered a minimum of 45% as a reasonable lower margin to address a mutation as germ-line. Consequently, mutations with MAFs < 45% were identified as somatic.(*)

Hayashi H, Kunimasa K, Tanishima S, Nakamura K, Ishikawa M, Kato Y, Aimono E, Kawano R, Nishihara H. Germline BRCA2 variant with low variant allele frequency detected in tumor-only comprehensive genomic profiling. Cancer Sci. 2024 Feb;115(2):682-686. doi: 10.1111/cas.16043. Epub 2023 Dec 12. PMID: 38086530; PMCID: PMC10859595.

6.-Authors should name the sequence of the primers or name the commercial primer set used.

Answer 6. Thank yhou for your observation with have added a paragraph about it:

Variants found to be of interest were Sanger sequenced for confirmation. Primers were designed with Primer3 (https://www.primer3.org) over the genomic sequences of the respective gene to acquire amplicons of size ranging from 150 to 250 bp. For PCRs, 1 µL of FFPE-extracted DNA, the Applied BiosystemsTM AmpliTaq GoldTM Master Mix (ThermoFisher Scientific, Vilnius, VA, Lithuania) and the corresponding forward and reverse designed primers (10µM) were used. PCRs were performed using the following thermal cycling conditions: an initial denaturation at 95ºC for 10 min, followed by 45 cycles of denaturation [94ºC, 45 seconds], annealing [temperature dependent primers melting temperature, 45 seconds] and extension [72ºC, 45 seconds], and a final extension at 72ºC for 5 mis Obtained amplicons were checked for correct size in 2.5% agarose gels at 90V. PCR products were cleaned-up with Exonuclease I (Takara, Sumilab, Kusatsu, Shiga, Japan) and Shrimp Alkaline Phospathase (Takara, Sumilab, Kusatsu, Shiga, Japan) in two steps: incubation [37ºC, 60 min] and inactivation [95ºC, 5 min]. Next, 2.5 µL of clean products were sequenced in both, the forward and reverse, directions using the BigDyeTM  Terminator v3.1 Cycle Sequencing Kit (Applied BiosystemsTM, ThermoFisher Scientific, Vilnius, VA, Lithuania) and forward or reverse primer (10µM). The thermal cycling conditions were: initial denaturation at 95ºC for 1 min, followed by 25 cycles of denaturation [90ºC, 30 seconds], annealing [temperature dependent primers melting temperature, 15 seconds] and extension [60ºC, 2 min]. Sequencing clean-up was performed with the Performa Dye Terminator Removal Gel Filtration Cartridges (Edge Biosystems Inc, ThermoFisher Scientific, Vilnius, VA, Lithuania) according to manufacturer’s instructions. For PCR amplification, PCR product clean-up and Cycle Sequencing, a Mastercycler® ep thermal cycler (Eppendorf) was used. Finally, capillary electrophoresis of marked samples was carried out in the genomics section of the central support service for experimental research of the University of Valencia. Chromatograms were manually aligned to the NGS reads to check for confirmation of the mutation.

Additionally, the reviewer will surely appreciate the table of the first primers designed for the study.

7.- A weakness of this statistical evaluation is the regression of the DNA data with the CA-125 data. Firstly, CA-125 is only a conditionally meaningful marker for ovarian cancer. Secondly, the r-value is unusably poor compared to TP53.

Answer 7: We agree that CA125 is the “gold standar” marker for ovarian cancer, so our aim with this regression analysis was to explore if both the cfDNA and TP53ctDNA measurements (and BRCA1ctDNA when available) werecorrelated with CA125 measurements.

First, the correlations are calculated separately for each patient. The table below, shows a summary of the values obtained:

Patient

Correlation cfDNA-CA125 (p-value)

Correlation TP53- CA125  (p-value)

Correlation BRCA1cfDNA-CA125 (p-value)

Table in the text

5

0.88 (p<0.001)

0.73 (p=0.01)

Table 3

11

0.87 (p<0.01)

0.43 (p=0.02)

0.52 (p=0.05)

Table 4

10

0.68 (p<0.001)

0.66 (p<0.001)

Table 5

8

0.61 (p=0.04)

0.71 (p=0.001)

Table 6

We get significant correlations in all the cases. So, despite the limited number of observations per patient, we get high correlations, especially between the cfDNA and CA125 measurements of each patient.

Secondly, as patients 5 and 11 have a similar clinical evolution and we have grouped them in a common group, we calculate a correlation coefficient for repeated measures, to check if they share a common correlation performance.  In this case, as stated in the text, “We obtained a positive and significant aggregated correlation between CA125 and cfDNA (r = 0.87; p = 0; Figure 5A) and a positive correlation very close to statistical significance between CA125 and TP53 ctDNA (r = 0.65; p = 0.058; Figure 5B).”

8.-For non-expert readers, a legend to Table 1 should be available explaining the abbreviations (e.g. PCI, or R Surgery).

Answer 8.-Thank you for your observation, we have added the description of acronyms and redone the table so that it is more understandable to the inexperienced reader.

Section A in Figure 3 is missing and has probably slipped into the text on page 9.

Answer : Thank you for your observation again , We have reviewed the layout because in the original the images are well placed

The title in Fig. 4 cannot be reassigned.

Already corrected

The designations A and B are missing in Fig. 5.

Already corrected

9.-The positive correlation in Fig. 5 B is to be rejected with an r-value of 0.65. The claim of a correlation is unscientific with these values.

We agree with the reviewer. As mentioned above, the correlation is not significant in this case. The p-value is 0.058, which is very close to the statistical significance, but does not reach values below 0.05. Note that this value has been obtained with only 2 patients. This encourages us to continue working to get a larger sample of patients with the same clinical evolution.

10.-The reviewer does not rule out that the results of this study provide a good hint of indicative markers that may be more effective than CA125 alone. How the authors discuss, this must also be discussed and requires a meaningful study.

Answer 10. Totally agree with the reviewer, This is a pilot study that outlines the future lines of research of our group. <Of course, a larger prospective study is required to validate all these initial results, we hope to launch it in the near future.

11.-The authors correctly state that previous successes in detecting tumor-relevant mutations are too expensive. The authors should it make more clear, that detecting specific mutations requires precise knowledge of what the clinician is looking for and a corresponding ddPCR test. This would be very time-consuming, as the authors themselves confirm the high variability of the mutations. The fact that this procedure is unlikely to lead to a broadly applicable detection test is the real statement that should be discussed with these data. To prove that cfDNA can indicate recurrences, as described here, larger studies are needed.  

Answer 11.

 We agree with you. Thanks for this appreciation.

A very wide variety of mutations are found in ovarian tumours and the sequencing technique is very expensive. However, we believe that if a mutation is determined in a tumour, it allows us to monitor it with the aim of identifying the recurrence of that tumour clone and adapting the treatment, especially in those patients in whom the mutations determine the use of maintenance treatments, as in the case of iPARP.

On page 8, lines 263 to 265, the usefulness of ddPCR for patient follow-up is highlighted.

In patient 5 there is evidence that the diagnostic mutation in TP 53 is elevated, being concordant with the elevation of cfDNA and with the appearance of recurrence, showing the usefulness of this technique in the mentioned context (Page 9; L 300 - 303).

Similarly, patient 11 has mutations in TP53 and BRCA, showing an increase in copies correlated with an increase in mutations in the BRCA gene. (Page 17, L 412 - 416)

We consider patient 8 to present the most surprising data, in which we are able to detect the recurrence by cfDNA, finding that the mutation we had previously was not the cause of this elevation, showing a different mutation in the hepatic recurrence in the recurrence compared to that found in the initial tumour. (Page 24; L525 - 531)

Based on the findings obtained in the sample, we consider that, although there is a high degree of heterogeneity in the mutations observed in ovarian cancer, knowing the predominant mutation in the tumour, it is possible to monitor and assess whether there are different mutations in the recurrence that allow or contraindicate the use of different drugs. (Page 27; 596 - 604)

Thanks for the suggestion.

The reviewer recognizes the importance of this study. However, this manuscript is only acceptable after a major, detailed revision.

I believe that I have made the most appropriate and effective corrections so that the readers of the journal will find this work interesting and useful for daily clinical practice. I hope you will consider our work again so that it can be published in your accredited journal.

We look forward to hearing from you at your earliest convenience.

Antoni Llueca MD PhD

Multidiciplinary Unit of Abdominal-Pelvic Oncology Surgery (MUAPOS)

University General Hospital of Castellon

Department of Medicine.

Medtronic University Chair for Training and Surgical Research

University Jaume I - UJI. Castellon. Spain

www.castellonschoolsurgery.com

Reviewer 2 Report

Comments and Suggestions for Authors

In general, the article is technically well written with a concise expression. The presented topic is of great interest for obstetricians and health care providers as Ovarian cancer treatment still represents a challenge. The use cfDNA and ctDNA represents a relative new method of monitoring of primary tumours and the metastatic disease burden.

My only concern regarding this study is the very small study population, thus I’m not sure about the statistical significance of the results, but I’m not in the position to comment the results. Maybe you can underline this important limitation as you mention only the “relatively brief follow-up”, maybe also detail again in this paragraph the total number of patients for each group etc.

I'd like to congratulate you for your study and encourage you to validate your findings through future studies with larger study populations.

Author Response

Prof. Dr. Andreas Kjaer

Editor-in-Chief

Diagnostics

Manuscript ID: diagnostics-3058250

Title: The Impact of Liquid Biopsy in Advanced Ovarian Cancer Care

10 July 2024

Dear Prof. Dr. Andreas Kjaer,

I have received your mail with the reviewer´s reports about our work. Attached please find the reviewers’ lined comments along with my comments explaining why I agree or disagree with their opinions.

Reviewer 2

In general, the article is technically well written with a concise expression. The presented topic is of great interest for obstetricians and health care providers as Ovarian cancer treatment still represents a challenge. The use cfDNA and ctDNA represents a relative new method of monitoring of primary tumours and the metastatic disease burden.

My only concern regarding this study is the very small study population, thus I’m not sure about the statistical significance of the results, but I’m not in the position to comment the results. Maybe you can underline this important limitation as you mention only the “relatively brief follow-up”, maybe also detail again in this paragraph the total number of patients for each group etc.

I'd like to congratulate you for your study and encourage you to validate your findings through future studies with larger study populations.

Answers:

Thank you very much for your comment, we have added a paragraph in the limitations section pointing out as a limitation the small number of cases that have been studied in the work

I believe that I have made the most appropriate and effective corrections so that the readers of the journal will find this work interesting and useful for daily clinical practice. I hope you will consider our work again so that it can be published in your accredited journal.

We look forward to hearing from you at your earliest convenience.

Yours sincerely,

Antoni Llueca MD PhD

Multidiciplinary Unit of Abdominal-Pelvic Oncology Surgery (MUAPOS)

University General Hospital of Castellon

Department of Medicine.

Medtronic University Chair for Training and Surgical Research

University Jaume I - UJI. Castellon. Spain

www.castellonschoolsurgery.com

Round 2

Reviewer 1 Report

Comments and Suggestions for Authors

The authors describe a method for the detection of cftDNA using qPCR and digital PCR, which they demonstrate in 22 patients with serous high-grade ovarian carcinoma. Analyses of the primary tumor were performed using FFPE material. The authors use this procedure as follow-up monitoring. Blood samples were taken after 1, 6, 9, 12 and 24 months. Finally, cfDNA was determined by qPCR and ctDNA by ddPCR and compared with CA125 level at seven patients. The authors actually recorded changes in ctDNA mutations in 83% of patients during the course of therapy. The reviewer considers the classification of patients into three groups based on the results to be rather risky due to the small number of cases. It is questionable how representative the significant correlations between TP53, BRCA and CA125 actually are. Because this is only shown in less cases. The authors defend the weak significance of the limited data and thus contradict the reviewer’s criticism instead of substantiating this with further findings from the 22 cases examined. Furthermore, the authors circumvent the reviewer's criticism and do not name the sequence of the primers. The reviewer's criticism of the misinterpretation of the regression lines in Figure 5 was replaced with direct significance tests. Nevertheless, the authors insist on the argument with the regression lines and show the invalid one twice in Figure 5, while the valid one has disappeared. Even the reviewer's suggestion that this test would provide better information when combined with CA125 and that this should at least be discussed is not made, despite the authors' consent.

I regret to say that too many aspects of the reviewer's work were not taken seriously enough. As a result, I am obliged to reject this manuscript in order not to jeopardize the quality of the journal.

Author Response

Academic Editor

Diagnostics

Manuscript ID: diagnostics-3058250

Title: The Impact of Liquid Biopsy in Advanced Ovarian Cancer Care

29 July 2024

Dear Academic Editor,

We would like to express our sincere gratitude for the comments and suggestions provided on our manuscript.

We have carefully considered all the comments and have made the necessary revisions.

Thank you once again for your guidance and support. We look forward to your further feedback.

Based on your suggestions, let us detail the most important changes.

  1. “Rethink the classification in three groups based on the results. It is hard to make a classification based on only one/two patients per group. Here the authors have two possibilities: 1) Increase the number of cases for each group or 2) remove the classification ad discuss the cases separately.”

Done. The classification has been removed and the cases have been discussed separately. The results section has been completely restructured.

  1. It would be important and also easy to provide the sequence of the primers. This will allow others to reproduce the experiments if and when necessary”.

 Thanks :Here are  the answer you required

The BIO-RAD (BioRad Laboratories,Hercules,CA,USA TM) company's mutation detection assays can be validated or designed on request by the researcher (customized). They are unique identification assays that you can request from BIO-RAD.

The company does not provide the primer and probe sequences under any circumstances.

The assays requested for this study are five, the first three validated and the last two not. They are the following:

  1. dHsaMDV2010105 for the TP53R175H mutation
  2. dHsaMDV2010127 for the TP53R248Q mutation
  3. dHsaMDV2510542 for the TP53G245D mutation
  4. dHsaMDS99673883 for the BRCA1 E272*(stop) mutation
  5. dHsaMDS544377207 for the TP53c.994C-1G mutation

  1. Please consider to meet reviewer 1 requests on figure 5 regression analysis”.

Done. We would like to apologise to the reviewer. We did not understand his request in the first review.

In order to clarify this analysis, the correlations between individualised cfDNA, ctDNA mutations and serum biomarkers have been grouped in a subsection. Information provided  from Tables 3,4,5 and 6 in the previous version of the paper has been combined into a single table  (Table 3 in the new version).

By eliminating the groups, the information given in Figure 5, which looked at the behaviour of patients (correlation between measurements over time) in one of the groups, has also been modified. The regression/correlation information in  Figure 5 (previous version) has been extended including more patients, as suggested by the reviewer.. This analysis is now explained in Figures 7,8 and 9, and the non-significant correlations have been commented in a more detailed way in the text.

  1. Discuss better the limitation of the study and the possibilities to improve its significance using also other circulating markers”.

Thanks for you observation with have rewrite the entire discussion in order to include your suggestions.

I believe that I have made the most appropriate and effective corrections so that the readers of the journal will find this work interesting and useful for daily clinical practice. I hope you will consider our work again so that it can be published in your accredited journal.

We look forward to hearing from you at your earliest convenience.

Yours sincerely,

Antoni Llueca MD PhD

Multidiciplinary Unit of Abdominal-Pelvic Oncology Surgery (MUAPOS)

University General Hospital of Castellon

Department of Medicine.

Medtronic University Chair for Training and Surgical Research

University Jaume I - UJI. Castellon. Spain

www.castellonschoolsurgery.com